# Comprehensive Detection of Known Attacks Using Integrated Datasets

Chaima Aouiche [+]
*Electronics and Telecommunications Department*
*Echahid Cheikh Larbi Tebessi University*
Tebessa, Algeria
chaima.aouiche@univ-tebessa.dz

Bolin Chen [+,*]
*Computer Science School*
*Northwestern Polytechnical University*
Xi'an, China
blchen@nwpu.edu.cn

Bairong Shen
*Institutes for Systems Genetics, West China Tianfu Hospital*
*Sichuan University*
Chengdu, China
bairong.shen@scu.edu.cn

Abdelaziz Aouiche
*Electronics and Telecommunications Department*
*Echahid Cheikh Larbi Tebessi University*
Tebessa, Algeria
abdelaziz.aouiche@univ-tebessa.dz

Rajeev K.Singla
*Department of Pharmacy and Institutes for Systems Genetics,West China Hospital*
*Sichuan University*
Chengdu, China
rajeevkumar@scu.edu.cn

Sahraoui Dhelim
*School of computing*
*Dublin City University*
Dublin, Ireland
sahraoui.dhelim@dcu.ie

* Corresponding Author: Bolin Chen
+ These authors have contributed equally to this work

*Abstract*—Cyberattacks, which are malicious attempts, are continuously increasing, leading to unauthorized data access, services disruptions, and network degradation. Efficient and proactive detection of these attacks is crucial to maintaining the confidentiality, integrity and availability of the digital environment. In this paper, we present an enhanced and comprehensive approach that cannot only detects known attacks but also identifies unknown ones through the integration of three up-to-date datasets and the implementation of sampling and feature selection techniques. To achieve this, we conducted experiments using two categories of methods: Machine Learning(ML), such as Naive Bayes (NB), Decision Trees (DT), Logistic Regression (LR), K-Nearest Neighbors (KNN), Random Forest (RF), XGBoost, and AdaBoost, and Deep Learning (DL) architectures, including Artificial Neural Networks (ANN), Deep Neural Networks (DNN), Convolutional Neural Networks (CNN), Long Short-Term Memory (LSTM), Gated Recurrent Units (GRU), and Recurrent Neural Networks (RNN). ML models offers easy interpretability, while DL models excel at handling complex patterns. The results from the majority of models show promising accuracy rates, with 99% for known attacks, significantly outperforming previous studies validating the effectiveness of our strategy.

*Index Terms*—Known Attack, Unknown Attack, Synthetic Oversampling, Feature Selection, Integrated Datasets

## I. INTRODUCTION

The rapid evolution of internet technologies, particularly with the advent of Industry 4.0 and the Internet of Things (IoT), has significantly transformed the digital landscape. These innovations have enabled high levels of connectivity, automation, and data exchange, thereby greatly enhancing daily life. However, the widespread adoption of these technologies has also created new vulnerabilities, which cybercriminals increasingly exploit [1]. This has led to more sophisticated and complex cyberattacks, escalating risks and posing significant threats to individuals, organizations, or governments alike. Consequently, the need for robust cybersecurity measures has never been more critical [2] .

Cyberattacks are generally defined as intentional and malicious attempts to damage, disrupt, or gain unauthorized access to digital systems and sensitive information [3], [4]. While these attacks can be categorized in various ways, we propose that the most practical and applicable approach is to classify them into two main groups: Known Attacks and Unknown Attacks. Known attacks are those that have been publicly disclosed and analyzed by cybersecurity experts. These attacks typically exploit well-documented vulnerabilities and use techniques that can be addressed through established security measures [5]. In contrast, Unknown attacks, also referred to

as Zero-Day Attacks, target vulnerabilities that are yet to be discovered by the public and cybersecurity experts, making them particularly dangerous. These attacks are characterized by their lack of detectable signatures in intrusion detection systems, allowing them to bypass traditional security mechanisms [6].

The diversity, dynamic nature and expansion of cyber threats make it a daunting challenge to identify, detect and defend against them. Traditional security solution such as encryption, access control and firewalls, while still necessary, have proven inadequate in identifying these threats [7], intrusion detection system (IDS) instead demonstrated to be critical to network security, particularly in detecting both known and unknown attacks [8]. In this context, Machine learning(ML) and Deep learning(DL)-based IDS methods have shown great promise in increasing detection accuracy and efficiency, thus overcoming the limitations of conventional methods that are no longer sufficient [5], [6], [9], [10]. Furthermore, the development of benchmarked network traffic based datasets has provided optimum options to accurately identify such attacks [8] [10]. For instance, Shahid et al. [11] used multiple ML algorithms, along with feature selection/dimensionality reduction techniques, to detect different attack classes, using only the CICIDS2017 dataset. Their study found that XGBoost was the best classifier with an accuracy of 99.91%. Ziadoon Kamil et al. [12] also applied both supervised and unsupervised ML algorithms to detect web attacks, showing that The KNN, DT and NB performed very well. Using the CSE-CIC-IDS2018 dataset, Gozde et al. [13] implemented six ML-based IDS coupled with the SMOTE oversampling technique, achieving a high accuracy rate. Similarly, Yung-Chung et al. [14] considered six DL models, with each model recorded accuracy levels above 98%. Recent research have leveraged the most renowned CIC-DDoS2019 datasets to rigorously evaluate their proposed methods. For example, Monika et al. [15] proposed three unsupervised learning algorithms with a GRP feature selection to detect zero-day DDoS attacks, obtaining an accuracy of 94.50%. In a similar effort, Mahrukh et al. [16] analyzed three DL models for both binary and multiclass DDoS attack classification reporting strong accuracy results.

Despite the impressive performance achieved by these studies, they may not fully reveal the evolving nature of cyber threats. Fortunately, the usage of combined datasets has been proved to be a powerful approach. In one study, Ali et al. [17] combined the CIC-IDS2017 and CIC-DDoS2019 to enhance the identification of GAN-adversarial DDoS attacks, resulting in a detection ratio between 91.75% and 100%. Another study by Mohamed Selim et al. [18] evaluated four ML algorithms on the CIC-IDS2017, CSE-CICIDS2018, and CIC-DDoS2019, both individually and in combination. Their approach, which utilized a broader and more varied training dataset, resulted in high accuracy for detecting IoV cyberattacks, along with reduced execution times.

Compared to the aforementioned studies, this paper aims to present an enhanced and comprehensive approach that can detects known attacks. Firstly, we integrate three up-to-

date intrusion datasets together, including:CIC-IDS2017, CSE-CICIDS2018, and CIC-DDoS2019. Secondly, after further preprocessing, we apply the Synthetic Over-sampling Technique (SMOTE) to address class imbalance in the combined dataset. For feature selection, we apply random forest (RF) algorithm, and we optimize hyperparameters using the Tree of Parzen Estimators (TPE). We then evaluate the performance of seven ML models  Naive Bayes (NB), Decision Trees (DT), Logistic Regression (LR), K-Nearest Neighbors (KNN), Random Forest (RF), XGBoost, and AdaBoost  and five DL models Artificial Neural Networks (ANN), Deep Neural Networks (DNN), Convolutional Neural Networks (CNN), Long Short-Term Memory (LSTM), Gated Recurrent Units (GRU), and Recurrent Neural Networks (RNN)).

The rest of this paper is organized as follows: Section 2 details the methods and materials used in this study. Section 3 presents and discusses the experimental results. Section 4 provides the conclusion.

## II. MATERIALS AND METHODS

### A. Materials

The development of this research followed a sequence of four main steps: datasets collection and preprocessing, features selection, models training, validation and evaluation. Additionally, class imbalance handling and parameters optimization, were performed as intermediate steps, as they are particularly important in improving model performance. Each of these steps is further explained in the following sections.

The datasets used in this study are: CIC-IDS2017, CSE-CICIDS2018, and CIC-DDoS2019, which collectively provide a comprehensive view of network flow information. These datasets are labelled and open-access developed by the Canadian Institute for Cybersecurity (CIC). They provide realistic network traffic scenarios for testing and evaluating intrusion detection and DDoS detection systems."TABLE. I" presents dataset specifications:

TABLE I: CIC datasets specifications

| Dataset Name | Samples | Features | Capture duration |
|---|---|---|---|
| CIC-IDS2017 | 3,119,345 | 85 | 5 days |
| CSE-CIC-IDS2018 | 16,232,943 | 80 | 10 days |
| CIC-DDoS2019 | 70,427,637 | 88 | 2 separate days |

### B. Method

*1) Data pre-processing:* Data preprocessing is an essential step in preparing datasets for model learning and testing. In this study, all three datasets were subjected to the same preprocessing operations. The following is a summary of the key preprocessing steps:

- **Initial Data cleaning:** First, the individual files within each dataset were merged separately: the eight files from the CIC-IDS2017 dataset, the ten files from the CSE-CIC-IDS2018 dataset, and the eighteen files from the CIC-DDoS2019 dataset. Following this, each merged dataset was then analyzed to detect and remove any

'NaN' values, 'INF' values, and duplicate entries, as well as repeated and unnecessary columns('Unnamed: 0', 'Flow ID', 'Source IP', 'Source Port','Destination IP', 'Destination Port', 'Timestamp' ). Once cleaned, the three datasets were combined, retaining only the common features, which were aligned in the same order across all datasets.

- **Labels Adjustment and encoding:** In the 'Label' column, Labels were standardized by combining similar entries, such as: 'benign' and 'BENIGN' into one 'BENIGN' category, 'UDP-lag' and 'UDPLag' into UDP-lag category. Labels representing very small data fractions were removed (Brute Force-XSS:230, SQL Injection: 87, DoS attacks-SlowHTTPTest:4, FTP-BruteForce:1). Then, after grouping the labels into 'benign' and 'malicious'(for all other attack types), they were encoded into numerical values, 0 for 'benign' and 1 for 'malicious' using label encoding.

*2) Class imbalance using SMOTE:* The Synthetic Minority Over-sampling Technique (SMOTE) is an effective method designed to address class imbalance by generating new synthetic samples for the minority class [19]. SMOTE creates new instances within the feature space by following these steps:

1) Nearest Neighbor Selection: For each minority class sample $x_i$, a nearest neighbor $x_{nn}$ is selected.
2) Difference Calculation: The difference between $x_i$ and $x_{nn}$ is computed:

$$diff = x_{nn} - x_i \tag{1}$$

3) Scaling the Difference: This difference is scaled by a random factor $r$ between 0 and 1:

$$gap = r * diff \tag{2}$$

4) Synthetic Sample Creation: The synthetic sample is generated as:

$$x_{syn} = x_i + gap = x_i + r * (x_{nn} - x_i) \tag{3}$$

Mathematically, for each minority class sample $x_i$, where the class label $y_i$=1, the synthetic sample $x_{syn}$ is generated as:

$$x_{syn} = x_i + random(0,1) * (x_{nn} - x_i) \tag{4}$$

In this study, despite a slight imbalance in the class distribution of newly retained combined-cleaned dataset(Malicious: 11885988, BENIGN:10836635), the SMOTE technique was successfully applied to the training set. This resulted in a balanced class distribution in the training set, with 8320492 instances in both classes (1:8320492, 0:8320492). The training and testing datasets were then normalized using Standard-Scaler for further subsequent analysis.

*3) Feature selection:* Feature selection is the process of identifying the most relevant features, which reduces execution time and improves model performance. There are several techniques available to perform this task including: mutual information, SHAP and principal component analysis with pearson correlation. For this study, feature selection was performed using a random forest classifier, which computed features importance weights [20]. Only the top 20 features with the highest importance score were retained after evaluating model performance with different feature counts. The selected features are presented in "Fig. 1".

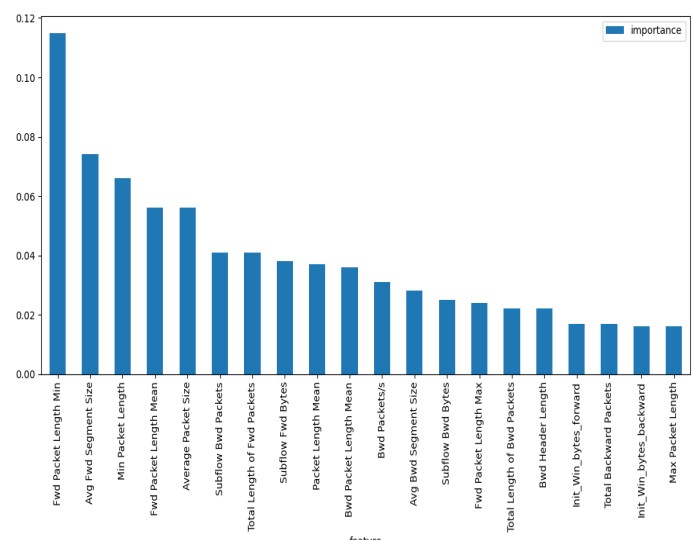

Fig. 1: Selected features of the combined dataset.

*4) Proposed ML and DL approaches training:* To identify known and unknown attacks seven ML and five DL algorithms were chosen. These selection were made due to their proven effectiveness in detecting attacks, as demonstrated in previous studies and discussed below [11]–[18]:

- ML algorithms are well-suited for handling complex and large datasets, identifying patterns in nonlinear data, improving model accuracy and processing speed.
- DL algorithms are affective at learning and adapting to complex, nonlinear, and long term patterns.

*5) Hyperparameter tuning:* Hyperparameter optimization is the process of adjusting a model's parameters to enhance its performance. This process aims to find the most effective combination of parameters that maximize the model's accuracy. Traditional methods like grid search and random search systematically explore the hyperparameter space, but these approaches can be computationally expensive. To address these challenges, this study utilizes the Tree of Parzen Estimators (TPE) algorithm, implemented through the Optuna framework. TPE [21] is an advanced Bayesian optimization method that uses probabilistic models to estimate the likelihood of different hyperparameters to select the best parameters. The optimal hyperparameter configurations obtained using 10 trials are listed below:

- DT:(`'max_depth':22`)
- RF:(`'n_estimators':72, 'max_depth':17`)
- LR:(`'C':0.7087354835479197`)
- NB:(`'n_estimators': 31,`
  `'learning_rate': 0.1902154150167555`)
- KNN:(`'n_neighbors':4`)
- AdaBoost:(`'n_estimators':31,`
  `'learning_rate':0.1902154150167555`)
- ANN:(`'units_1':17,`
  `'dropout_1':0.4439729818717545,`
  `'units_2':49, 'dropout_2':`
  `0.23449575971279818,'batch_size':53`)
- DNN:(`'units_1':160,`
  `'dropout_1':0.4494408400849417,`
  `'units_2':181,'dropout_2':`
  `0.23933926202890243, 'batch_size':53`)
- CNN:(`'filters':95,'units':106,`
  `'dropout':0.4260842730252883,`
  `'batch_size':75`)
- RNN:(`'units':142,`
  `'dropout':0.479401276081909,`
  `'batch_size':49`)
- LSTM:(`'units':194,`
  `'dropout':0.25984484810887,`
  `'batch_size':49`)
- GRU:(`'units': 170, 'dropout':`
  `0.26761846186661054,`
  `'batch_size': 81`)

*6) **Models validation and evaluation**:* To verify the accuracy of the models, k fold cross-validation was applied. Cross-validation (CV) is a robust technique used to assess the performance and generalizability of machine learning models. It involves dividing the dataset into multiple subsets, or folds, and systematically training the model on some of these folds while validating it on the remaining ones. For k=5, the dataset is split into 5 folds. The model is trained on four folds and validated on the fifth fold. This process is repeated five times, with each fold serving as the validation set once. The results are then averaged to provide a more reliable estimate of the model's overall performance.

## C. Results

In this section, we evaluate the effectiveness of our enhanced approach by presenting results using key performance metrics: accuracy, F1 score, precision, and recall. These metrics provide a clear and insightful understanding of how well our method performs.

*1) **Optimal hyperparameter models accuracy**:* The accuracy of ML/DL models using the optimal hyperparameter values determined by TPE is presented in"TABLE. II". As clearly observed, expect for NB(0.65), AdaBoost(0.96) and GRU(0,99), there was only a slight difference in the models accuracy before and after hyperparameters optimization. This indicates that hyperparameter tuning may not be so important for these models, as they perform with their default parameters.

TABLE II: Known Attacks classification models Accuracy after hyperparameter optimization

| Classifier | Accuracy |
|---|---|
| DT | 0.9917 |
| RF | 0.9939 |
| LR | 0.9110 |
| NB | 0.6689 |
| KNN | 0.9927 |
| AdaBoost | 0.8992 |
| XGBoost | 0.9943 |
| ANN | 0.9648 |
| DNN | 0.9800 |
| CNN | 0.9798 |
| RNN | 0.9789 |
| LSTM | 0.9810 |
| GRU | 0.9798 |

*2) **Model's 5-CV accuracy results**:* As illustrated in "TABLE. III" and "Fig. 2", all models, except NB, achieved the best performance with minimal standard deviation. Notably, RF, XGBoost, DT and KNN exceeded 99% accuracy, followed closely by LSTM, GRU, DNN and CNN.

TABLE III: Known Attacks detection 5-CV Accuracy results

| Classifier | Accuracy(5-CV) | Mean Accuracy | Standard Deviation |
|---|---|---|---|
| DT | [0.99424313 0.99412896 0.99426417 0.99372332 0.99418603] | 0.9941 | 0.0002 |
| RF | [0.99478397 0.99477195 0.99488612 0.9944349 0.9948831 ] | 0.9948 | 0.0001 |
| LR | [0.91557298 0.91494802 0.91595757 0.91478551 0.91525724] | 0.9153 | 0.0004 |
| NB | [0.65049576 0.64797488 0.65060693 0.64872198 0.6708391 ] | 0.6537 | 0.0086 |
| KNN | [0.99416201 0.99405985 0.99422811 0.99377139 0.99427316] | 0.9941 | 0.0002 |
| AdaBoost | [0.90412235 0.90359353 0.90438976 0.90297729 0.90314255] | 0.9036 | 0.0005 |
| XGBoost | [0.99444745 0.99446247 0.99446247 0.99414096 0.99448649] | 0.9944 | 0.0001 |
| ANN | [0.97352022 0.95948861 0.97368848 0.9736223 0.96840024] | 0.9697 | 0.0055 |
| DNN | [0.98248603 0.9820143 0.98265429 0.9841836 0.98248297] | 0.9828 | 0.0007 |
| CNN | [0.98290968 0.98158464 0.98456223 0.98392219 0.98092056] | 0.9828 | 0.0014 |
| RNN | [0.97427739 0.96934078 0.95419446 0.9798419 0.97680721] | 0.9709 | 0.0090 |
| LSTM | [0.98608858 0.98683973 0.98551469 0.98669547 0.9869749 ] | 0.9864 | 0.0005 |
| GRU | [0.98661739 0.98665044 0.98239589 0.98357065 0.98667744] | 0.9852 | 0.0018 |

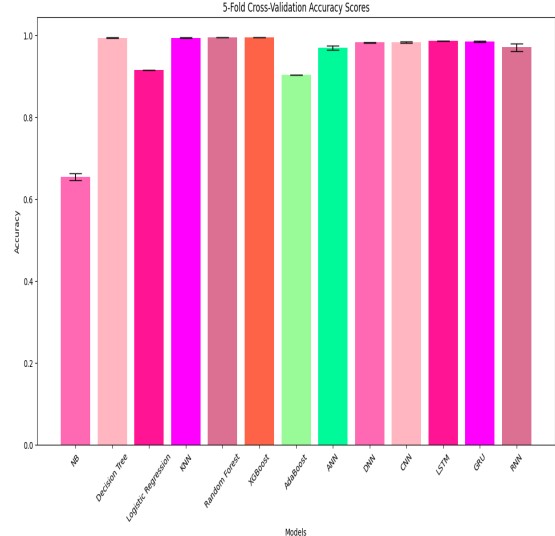

Fig. 2: Known Attacks 5-CV Accuracy scores.

*3) **ML and DL Models Evaluation**:* ML and DL algorithms were evaluated using the unseen test set. The results obtained are presented in "TABLE. IV" and shown in the confusion matrix depicted in "Fig. 3". The matrices indicate

that all models were able to correctly classify a significant number of instances. High true positive (TP) and true negative (TN) counts confirmed the ability of the selected models to detect both normal and known attacks. In addition, the highest accuracy, f1-score, precision and recall of RF, XGboost, KNN, GRU and AdaBoost further demonstrated their detection capabilities. NB is the worst classifier suffer the most from overfitting.

TABLE IV: Known attacks detection related models Evaluation

| Model | Precision | | Recall | | F1-Score | | Accuracy |
|---|---|---|---|---|---|---|---|
| | 0 | 1 | 0 | 1 | 0 | 1 | |
| DT | 0.9900 | 0.9976 | 0.9974 | 0.9909 | 0.9937 | 0.9942 | 0.9940 |
| RF | 0.9896 | 0.9991 | 0.9990 | 0.9904 | 0.9943 | 0.9947 | 0.9945 |
| LR | 0.8519 | 0.9847 | 0.9856 | 0.8438 | 0.9139 | 0.9088 | 0.9114 |
| NB | 0.9601 | 0.6130 | 0.3160 | 0.9880 | 0.4755 | 0.7566 | 0.6675 |
| KNN | 0.9886 | 0.9988 | 0.9987 | 0.9895 | 0.9937 | 0.9942 | 0.9939 |
| AdaBoost | 0.8265 | 0.9980 | 0.9983 | 0.8089 | 0.9043 | 0.8936 | 0.8992 |
| XGBoost | 0.9890 | 0.9992 | 0.9992 | 0.9899 | 0.9941 | 0.9945 | 0.9943 |
| ANN | 0.9231 | 0.9960 | 0.9960 | 0.9243 | 0.9581 | 0.9588 | 0.9585 |
| DNN | 0.9695 | 0.9940 | 0.9936 | 0.9715 | 0.9814 | 0.9827 | 0.9821 |
| CNN | 0.9703 | 0.9950 | 0.9946 | 0.9722 | 0.9823 | 0.9835 | 0.9829 |
| LSTM | 0.9762 | 0.9954 | 0.9951 | 0.9779 | 0.9856 | 0.9866 | 0.9861 |
| GRU | 0.9709 | 0.9985 | 0.9984 | 0.9727 | 0.9845 | 0.9854 | 0.9850 |
| RNN | 0.9661 | 0.9808 | 0.9792 | 0.9687 | 0.9726 | 0.9747 | 0.9737 |

## D. Conclusion

The mechanism of attacks detection is too complex to be revealed by single dataset. Combining multiple datasets can be leveraged to identify more sophisticated attacks. Recent up-to date datasets represent a powerful source of information about known and new attacks-long term evolution. Our findings underscore the importance of using such datasets in information security tasks including intrusion detections. Our main objective in this paper was not only the detection of known attacks but also unknown ones by excluding four specific attack types (NTP, DNS, SNMP, SSDP and TFTP) from the combined datasets, reserving them for model evaluations. These attack types are existed in CIC-DDoS-2019 day2 but not in day 1. Due to time limitations, we were able to detect only known attacks.

## E. Authors contributions

CA initialized this study. BC and CA discussed many times to finalize the work plan. BS gave suggestions many times to modify this study. BC and CA conducted the majority of numerical experiments. CA, BC, AO, RK.S and SD drafted the manuscript. Everyone read the manuscript and revised it, and agreed with the final version.

## F. Funding

This work was supported by the National Natural Science Foundation of China under Grant No. 62433016, 61972320.

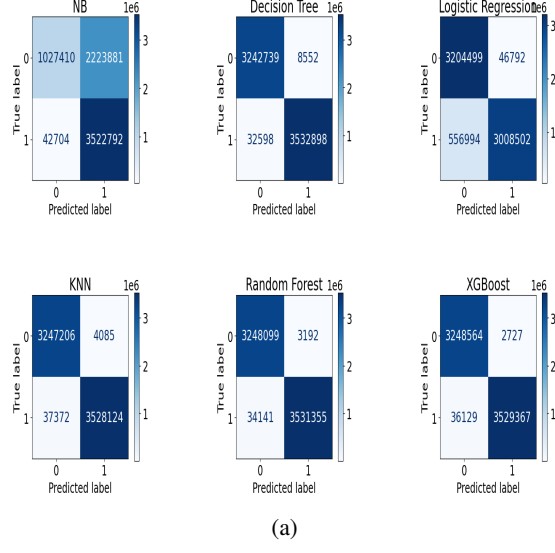

(a)

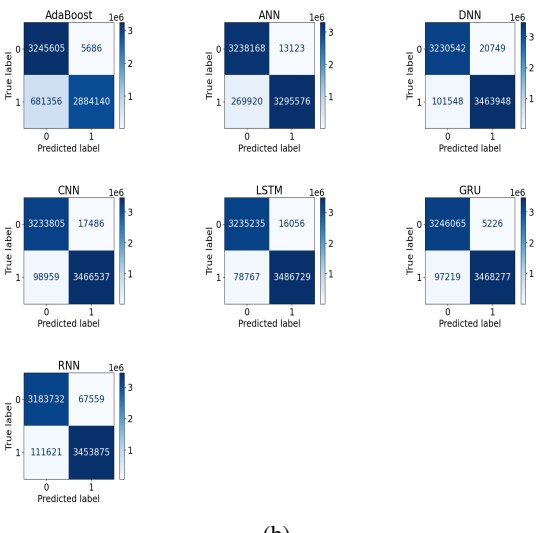

(b)

Fig. 3: Known Attacks detection confusion matrix.

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
