# OpenReview forum: "Comprehensive Detection of Known Attacks Using Integrated Datasets"
_IEEE.org/ICIST/2024/Conference — IEEE ICIST 2024 Conference Submission_

### Official Review · Reviewer_scah · 2024-08-21
**This manuscript has a certain degree of innovation and clear simulation figures. It is recommended to accept this paper for publication in IEEE ICIST 2024.**

**Rating:** 7
**Confidence:** 4

**Review:**

This manuscript has a certain degree of innovation and clear simulation figures. Please answer the following review questions.

The paper mentions the integration of three up-to-date datasets and the use of sampling and feature selection techniques. Could you elaborate on the specific reasons why these datasets were chosen and how they contribute to the detection of both known and unknown attacks?

How robust is the proposed approach to detecting unknown attacks, which are often more challenging to identify? What measures were taken to ensure the generalizability of the results to other datasets or attack scenarios?

---

### Official Review · Reviewer_Aba8 · 2024-08-21
**This paper presents an enhanced and comprehensive approach. The paper is well written. Here are some comments.**

**Rating:** 7
**Confidence:** 3

**Review:**

Question 1:
Please elaborate on how the integration of the three datasets and the implemented sampling and feature selection techniques contribute to the detection of both known and unknown cyberattacks. What specific challenges were addressed through these techniques?
Question 2:
How do the Machine Learning (ML) models and Deep Learning (DL) architectures compare in terms of their ability to detect known and unknown attacks? Are there specific scenarios or types of attacks where one approach significantly outperforms the other?
Question 3:
The paper reports a 99% accuracy rate for known attacks. Please provide more details on the validation process used to achieve this accuracy. How does this performance compare with state-of-the-art methods in terms of detection rate and false positives?

---

### Official Review · Reviewer_Jy6S · 2024-08-21
**Manuscript Accept**

**Rating:** 7
**Confidence:** 4

**Review:**

How are the known attacks identified?
What are the concrete contents of three up-to-date datasets?
How do the authors guarantee the universality of the proposed detection mechanism for different attacks?

---

### Decision · Program_Chairs · 2024-09-08

Accept (Oral)